# Role of Dietary Flavonoids in Iron Homeostasis

**DOI:** 10.3390/ph12030119

**Published:** 2019-08-08

**Authors:** Marija Lesjak, Surjit K. S. Srai

**Affiliations:** 1Department of Chemistry, Biochemistry and Environmental Protection, Faculty of Sciences, University of Novi Sad, Trg Dositeja Obradovića 3, 21000 Novi Sad, Serbia; 2Research Department of Structural and Molecular Biology, Division of Biosciences, University College London, Darwin Building, Gower Street, London WC1E 6BT, UK

**Keywords:** iron homeostasis, iron absorption, non-haem iron, flavonoids

## Abstract

Balancing systemic iron levels within narrow limits is critical for human health, as both iron deficiency and overload lead to serious disorders. There are no known physiologically controlled pathways to eliminate iron from the body and therefore iron homeostasis is maintained by modifying dietary iron absorption. Several dietary factors, such as flavonoids, are known to greatly affect iron absorption. Recent evidence suggests that flavonoids can affect iron status by regulating expression and activity of proteins involved the systemic regulation of iron metabolism and iron absorption. We provide an overview of the links between different dietary flavonoids and iron homeostasis together with the mechanism of flavonoids effect on iron metabolism. In addition, we also discuss the clinical relevance of state-of-the-art knowledge regarding therapeutic potential that flavonoids may have for conditions that are low in iron such as anaemia or iron overload diseases.

## 1. Biological Importance of Iron

Studying the chemistry of iron in detail, it is easy to see why iron is essential for life. Namely, under physiological conditions, iron is mainly present in two forms, ferrous (Fe^2+^) and ferric (Fe^3+^). The Fe^3+^ Fe^2+^ system facilitates variety of redox potentials that can be fine adjusted by different ligands (from about −0.5 V to about +0.6 V), which almost entirely corresponds to the redox potential range of utmost importance for biological systems. That is why iron complexes are uniquely suitable for a variety of catalytic processes and reactions which are of great biological significance, such as electron transfer and acid-base reactions [1,2].

Iron carries out a variety of significant roles in biological systems, mostly as a part of iron-containing proteins. Haemoproteins are a large group of iron-containing proteins where the iron is bound to a porphyrin molecule (haem) which is bound to the different proteins with diverse functions. There are three main categories of haem proteins: Oxygen carriers (haemoglobins, myoglobins and neuroglobins), activators of molecular oxygen (cytochrome oxidase, cytochrome P450s, catalases and peroxidases) and electron transport proteins (cytochromes) [3,4].

The second group of iron-containing proteins is the iron-sulphur proteins, where iron is bound to sulphur by thiol groups from cysteine or inorganic sulphide. Iron-sulphur proteins are widespread in all living organisms and express numerous actions. Namely, they are included in redox and non-redox reactions as part of different enzymes, like succinate dehydrogenase and aconitase, and proteins involved in the electron transfer chain [1,5].

The third class of iron-containing proteins presents a diverse group of proteins that do not contain iron in a haem or iron-sulphur form. One group is mononuclear non-haem iron enzymes, which include lipoxygenases, aromatic amino-acid hydroxylases, prolyl and lysyl hydroxylases, etc. Additionally, there is the dinuclear non-haem iron protein group, consisting of ribonucleotide reductase and ferritins or proteins involved in iron transport, such as transferrins (Tfs) [1].

Summing activities of the above-mentioned proteins, it is apparent that iron is crucial for many important processes, such as: Oxygen transport and storage, cellular respiration and energy production, the electron transport chain of mitochondria, synthesis of DNA, RNA and proteins, regulation of gene expression, cell proliferation and differentiation. In addition, iron is indispensable for normal brain function, psychomotor development and cognitive performance (especially in infants), endurance and physical performance, the inflammatory response, pregnancy (40% of all maternal prenatal deaths are linked to anaemia), thyroid function, production and metabolism of catecholamines and other neurotransmitters, drug metabolism, etc. Hence it is evident that nearly every cell and organism require iron for life [3,6,7,8,9,10].

On the other hand, the property of iron to easily change its oxidative state can also be toxic, mainly due to its ability to produce free radicals when it is not bound by proteins and is free in a labile iron pool. Iron takes part in a reaction, known as the Fenton reaction, where the hydroxyl radical (HO^•^) is the end product. HO^•^ is the most toxic reactive oxygen species (ROS) which can damage all classes of biomolecules. Consequently, unrestrained production of HO^•^ leads to cell injuries and death and gives rise to numerous severe pathological states [11]. The Fenton reaction initiates the chain reaction (Equation (1)), which is then followed by the reactions (Equations (2) and (3)) in which more and more HO^•^ is produced [12].
Fe^2+^ + H_2_O_2_ → Fe^3+^ + HO^•^ + HO^−^(1)
HO^•^ + H_2_O_2_ → H_2_O + O_2_^•−^ + H^+^(2)
O_2_^•−^ + H^+^ + H_2_O_2_ → O_2_ + HO^•^ + H_2_O(3)

Thus, balancing systemic iron levels within narrow limits in an organism is crucial, as both iron deficiency and iron overload lead to serious haematological, metabolic and neurodegenerative disorders, which belong to the most frequent disorders worldwide, as well as carcinogenesis [13]. 

## 2. Distribution and Homeostasis of Body Iron

The total iron content of the adult human organism is estimated around 4 g (~35 mg/kg woman, ~45 mg/kg for men). About 66% of total body iron is found as part of haemoglobin in circulating erythrocytes, erythrocyte precursors or as intracellular pool (liver and reticulo-endothelial macrophages), 7.5% in muscle as part of myoglobin, 0.5% as part of the catalytic center of a variety of enzymes (cytochromes, catalase, peroxidases, flavoproteins, etc.) and 0.1% as Tf-bound iron in the circulation (see Figure 1) [2,14].

Body iron homeostasis is maintained by regulating the iron levels in plasma (Tf-bound iron), which is determined by four coordinated processes: Duodenal iron absorption, macrophage iron recycling, hepatic iron storage and erythropoiesis. Erythropoiesis, the production of red blood cells in bone marrow, requires nearly 30 mg iron each day, the main part of which comes from the recycling of iron via reticulo-endothelial macrophages (>28 mg/day). Macrophages ingest old or damaged erythrocytes, process them and release recycled iron to plasma Tf. The pool of Tf-bound iron (~3 mg) is very dynamic and undergoes recycling more than 10 times daily. Furthermore, when in balance, each day the body absorbs 1–2 mg of iron by duodenal enterocytes and at the same time loses 1–2 mg of iron by nonspecific iron losses, such as exfoliation of enterocyte, skin and hair loss, menstruation and some gastrointestinal blood loss (Figure 1). Bearing in mind that there is no known physiologic mechanism for controlling iron excretion and that macrophage-mediated iron recycling cannot be sufficient for maintaining erythropoiesis over the long term, absorption of dietary iron in duodenum is of great importance in keeping iron homeostasis in balance [15,16].

### Mechanism of Dietary Iron Uptake

Nutritional iron absorption occurs primarily in the duodenum, on the apical (luminal) membrane of the enterocytes, and is tightly regulated by bioavailable iron, iron stores, erythropoietic drive and inflammation. The average diet daily contains about 10–20 mg of iron from which only 1–2 mg is absorbed. There are two types of dietary iron: Non-haem iron, which is present in food from both animal or plant origin, and haem iron, which is present only in food of animal origin. Absorption of non-haem iron in the intestine comprises the following (see Figure 2) [17]:Reduction of Fe^3+^ and uptake of Fe^2+^ from the diet through the apical membrane of enterocytes. In the diet iron is mainly present as Fe^3+^. However, the absorption of Fe^2+^ is more efficient than Fe^3+^. In order to increase Fe^3+^ bioavailability, Fe^3+^ firstly needs to be reduced. Duodenal cytohrome b (Dcytb) is an iron-regulated ferric reductase, highly expressed on the apical membrane of duodenal enterocytes [18]. After being reduced by Dcytb, Fe^2+^ is transported across the apical membrane by the divalent metal transporter 1 (DMT1) [19].Intracellular processing of iron and iron transport to the basolateral membrane of enterocytes. Even though mechanisms of intracellular iron transport are not fully elucidated, it is assumed that poly r(C)-binding proteins (PCBPs) play important roles in this transport. Namely, PCBP1 is identified as an iron chaperone for ferritin, the main iron storage protein in the cell, while PCBP2 is assumed to transfer of iron from DMT1 to the cytosol and later to iron efflux transporter ferroportin (FPN). In addition, NCOA4 was identified as autophagic receptor for ferritin, which during iron deficiency in cell leads to ferritin autophagy and iron liberation [20]. In general, the fate of absorbed iron is closely related to the body’s demands for iron. If there is a need for more iron, then iron is exported from the cell via the basolateral membrane of enterocytes which is followed by iron binding to Tf and transport to peripheral tissues that require iron. If there is no need for additional iron in the body, iron is stored in the cell in the form of ferritin, and returned to the lumen at a time when the villus enterocytes die [8].Transfer of iron through the basolateral membrane to the circulation. The mechanism of Fe^2+^ transport through the basolateral membrane includes synchronized activity of two proteins, FPN [21,22,23] and transmembrane copper-dependent ferroxidase, hephaestin (Heph) [24,25]. Before entering circulation, Fe^2+^ firstly needs to be oxidized to the Fe^3+^ state, which is catalysed by hephaestin, the intestinal ferroxidase. Fe^3+^ then binds to the serum glycoprotein Tf [26], the key iron transporting protein in the serum and extracellular fluids.

The uptake mechanism for non-haem and haem iron differs across the apical membrane of the enterocyte, while it follows the same pathway once iron is inside the cell (see Figure 2). Even though the mechanism of haem absorption is not fully characterized, haem carrier protein 1 (HCP1) was identified as protein for haem uptake on the apical membrane of duodenal enterocytes [27]. Currently, the role of HCP1 in haem transport is debated since it was also identified as the proton-coupled folate transporter [28]. In addition, the new heam transporter is identified, known as feline leukemia virus subgroup C cellular receptor family member 2 (FLVSC2), whose detailed characterisation is in process [29]. Once inside the cell, haem is degraded by haem oxygenase (HO-1) [30] and the released iron enters an intracellular iron pool. After that, absorbed iron from the haem source follows the pathway of absorbed non-haem iron.

## 3. Bioavailability of Iron

To keep iron in balance, it is essential that iron is supplied by diet, especially during growth of infants, children and adolescents and the reproductive period in women, particularly during pregnancy. In Table 1 it can be clearly seen that daily requirements of absorbed iron differ greatly between individuals of different age, sex and state [31,32].

As mentioned previously, dietary iron occurs in two forms: Haem and non-haem. Haem iron makes 10–15% of total iron from diet in meat-eating populations, but it is estimated to contribute ≥40% of total absorbed iron. However, non-haem iron absorption is much lower, and it varies between 2–20%. In contrast to non-haem iron, whose bioavailability is highly dependent on the presence of iron absorption promoters or inhibitors in the diet, dietary factors have little effect on haem iron absorption [31].

### 3.1. Anaemias

The low bioavailability of non-haem iron contributes greatly to iron deficiency anaemia (IDA), which is the most prevalent nutritional deficiency worldwide, estimated to affect two billion people especially in low-income populations where consumption of meat is low. On the other hand, low bioavailability of non-haem iron is a problem in population groups eating only a plant-based diet, vegetarians and vegans, whose popularity is rising in modern societies [34].

Iron deficiency disorders are generally known as functional deficiency and anaemias. Commonly, anaemia is a condition in which there are not enough healthy erythrocytes in the circulation which leads to inadequate oxygen distribution and consequently disturbance in the maintenance of normal physiological function of tissues, such as liver, brain, muscles, etc. [33].There are many types of anaemia and these can arise as a result of a wide variety of causes that can be single, but more often coexist. Globally, the most significant contributor to the anaemia is IDA. The main causes for IDA are low dietary iron intake, poor absorption of iron from diet at a period of life when iron requirements are particularly high, such as growth periods among children, reproductive period among women, especially during pregnancy. Other recognized causes of anaemia, such as heavy blood loss, extensive menstruation or chronic bleeding are also recognized [35].

Additionally, anaemia and hypoferraemia that occurs as consequence of chronic infections and inflammatory disorders is known as anaemia of inflammation (AI) or anaemia of chronic disease. AI is a systemic iron disorder characterized with decreased iron, iron binding capacity and intestinal iron absorption, as well as impaired erythropoiesis, while iron is trapped in macrophages and liver, indicating impaired mobilization of iron from stores. AI is a consequence of cytokine (mainly IL-6) mediated induction of hepcidin production as a response to chronic inflammation [3,36].

To be more precise, anaemia is a consequence of both poor nutrition and poor health. Increased risk of maternal and child mortality is one of the main concerns of severe anaemia. Additionally, the negative consequences of IDA on cognitive and physical development of infants and on general performance, particularly work productivity in adults, are also great concern. The World Health Organization declares iron deficiency as one of the 10 leading risk factors for disease, disability and death in the world today. Iron deficiency affects mostly children and women in practically all countries. It can be estimated that most preschool children in non-developed countries and at least 30–40% in developed countries are iron-deficient, and nearly half of the pregnant women in the world are estimated to be anaemic [6,34,35,37,38].

In order to compensate for lost iron and to keep iron homeostasis in balance, it is of utmost importance that absorption of iron is sufficient. Thus, it is essential to understand in detail the mechanism of iron absorption in the duodenum as well as to target its promoters or inhibitors. Additionally, for individuals affected with iron deficiency (anaemias) it is important to know what food is rich in highly bioavailable iron and try to consume it as much as possible. In Table 2 it can be seen what the average levels of total iron in common foods are.

### 3.2. Dietary Inhibitors of Iron Absorption

There are recognized inhibitors of iron absorption whose occurrence in food should be addressed in iron-deficient individuals. Major inhibitors of iron absorption from the diet are phytate, polyphenols (especially flavonoids), calcium and proteins.

Phytate (inositol hexakisphosphate; see Figure 3) is a primary phosphorous storage molecule in plants and cannot be digested by humans.

It is believed that phytate forms a complex with iron through its phosphate ester groups making it nonabsorbable and it is considered as the main inhibitor of non-haem iron absorption. The inhibitory effect of phytate has been proven, but particular food preparation methods, such as milling, heat treatment, soaking, germination, fermentation, addition of ascorbic acid or enzyme phytase, can remove or degrade phytate and thus partially or totally eliminate its negative effect on non-haem iron absorption. However, low concentrations of phytate (2–10 mg/meal) express a negative effect on non-haem iron absorption [31,40,41,42]. Considering that some foods contain phytate in considerable concentrations, even much more than common food containing non-haem iron (see Table 3), consumption of phytate rich plants should be under attention, especially together with iron supplementation. Plant foods not shown in Table 3 may be considered as not rich in phytate.

Calcium has been shown to have an inhibitory effect on both non-haem and haem iron absorption. The mechanism of the inhibitory effect of calcium on iron absorption is not known, but it is speculated that it could block initial iron uptake by the enterocyte [44]. Inhibition of iron absorption has been demonstrated even with a calcium concentration that is common in the daily dietary intake. This fact could represent a general health problem because widespread and recommended use of calcium supplements, manly for prevention of osteoporosis, can bring about problem with iron absorption [31,45].

Particular proteins are also proven to have an inhibitory effect on iron absorption such as: Milk, soybean and egg proteins, albumin, casein and whey [46,47,48,49].

### 3.3. Dietary Enhancers of Iron Absorption

The main dietary enhancers of iron absorption are ascorbic acid and muscle tissue. It is proven that ascorbic acid improves non-haem iron absorption, mainly due to its ability to reduce Fe^3+^ to Fe^2+^ and thus make it available for transport by DMT1. The amount of ascorbic acid that expresses a positive effect on non-haem iron absorption is approximately 30–100 mg daily, which corresponds to the recommended dietary intake for ascorbic acid. However, in foods of plant origin, such as, fruits and vegetables, the supporting effect of ascorbic acid might be reduced by the inhibiting effect of polyphenols and phytate [40,50,51,52,53,54]. In contrast to the positive effect of ascorbic acid on non-haem iron absorption after a single meal, improvement in iron status after chronic supplementation with ascorbic acid was not observed in humans. The reason for this occurrence is not yet fully understood [55].

Muscle tissue, known as the “meat factor”, also showed a positive effect on non-haem iron absorption, the same as ascorbic acid, but it was hard to demonstrate the same activity after a longer consumption. There is evidence that this could be attributed to: Cysteine-containing peptides, glycosaminoglycans and L-α-glycerophosphocholine and their ability to reduce and chelate iron [56,57,58].

### 3.4. Ways to Prevent Anaemia

Nowadays three approaches are recognized as ways to deal with IDA and raise amount of absorbed iron which can be practices alone or in combination with each other: Change in dietary habits by means of diversity and modification of the diet in order to improve nutritional value andiron bioavailability, supplementation (intake of iron in higher doses not with food), and fortification (the addition of iron into food during food processing).

A change of dietary habits so that intake of food rich in both haem and non-haem iron, as well as promoters of iron absorption, is increased, while intake of inhibitors of iron uptake should be decreased. Even though it showed significant practical limitations, a change of dietary habits is the favoured way of treating IDA. Apart from the fact that it is hard to change an individuals’ dietary preference, food rich in highly bioavailable iron, such as meat, is expensive especially in developing countries.

Supplementation is an efficient and cost-effective way of treating IDA over short periods of time, such as pregnancy. However, insufficient coverage of all parts of the world and compliance is a major limitation to the effectiveness of iron supplementation programs [59]. Iron supplementation is carried out orally or, very rarely, by injection. Frequently used forms of iron in supplements include Fe^2+^ and Fe^3+^ salts, such as SO_4_^2−^, gluconate, fumarate and citrate. High doses of supplemental iron may cause gastrointestinal side effects, such as nausea and constipation. Other forms of supplemental iron, such as haem iron, carbonyl iron, iron amino-acid chelates and polysaccharide-iron complexes, are also available and are believed to manifest fewer gastrointestinal side effects compared with salts [6,60].

Iron fortification of food is considered as the most cost-effective route for lowering incidence of IDA all over the world. Generally, iron fortification refers to the addition of iron to foods consumed by all or most of the population and it is regulated by the government. Milled cereals are frequently the subject of iron fortification and showed a successful outcome in making populations less iron deficient. In addition, it was estimated that iron fortification is economically more favourable than iron supplementation [60,61].

## 4. Plant Polyphenols

Polyphenols are plant secondary metabolites that include a great number of structurally diverse compounds. Chemically speaking, phenols are compounds which contain one (phenol) or more (polyphenols) aromatic rings, bearing one or more hydroxyl groups, which can be esterified, etherified or glycosylated. Generally, polyphenols represent all secondary metabolites whose syntheses go through the shikimate/phenylpropanoid or the “polyketide“ acetate/malonate pathway, or by combination of two of them, producing monomeric or polymeric phenols. Additionally, phenols are uncommon in bacteria, fungi and algae but are ubiquitously present in the plant kingdom. The phenolic profile of an individual plant strongly depends on plant species and thus can be used as a reliable taxonomic marker [62]. Throughout evolution, plants have developed adaptive mechanisms which are reflected in their ability to produce a great number of phenolic secondary metabolites. Although phenols are not compulsory in the processes such as plant growth and development, they have pivotal role for plants’ interactions with the environment, reproduction and defence. From an evolutionary point of view, it is easy to see why plants produce such a great collection of secondary compounds compared with animals. Namely, they cannot rely on physical mobility to escape predators or perform successful pollination. Thus, they had to developed exuberant chemical systems in order to survive. Plants need phenols for protection against herbivores, microbes, viruses or other plants, as signal compounds to attract pollinating or seed dispersing animals, protection from ultraviolet radiation or oxidants and fluctuation of organic and inorganic nutrients from soil [62].

Phenols are generally soluble in polar organic solvents, unless being entirely esterified, etherified or glycosylated. Also, most phenol glycosides are water-soluble but the corresponding aglycones are usually less so. Due to the presence of an aromatic ring, all phenols demonstrate intense absorption in the ultraviolet part of the spectrum. Furthermore, phenols that give colour to plants absorb light in the visible region as well. On the basis of the phenol skeleton, several classes of phenols have been categorized: C_6_ (phenols, benzoquinones), C_6_–C_1_ (phenolic acids), C_6_–C_2_ (acetophenones, phenylacetic acids), C_6_–C_3_ (hydroxycinnamic acids, coumarins, phenylpropanes, chromones), C_6_–C_4_ (naphthoquinones), C_6_–C_1_–C_6_ (xanthones), C_6_–C_2_–C_6_ (stilbenes, anthraquinones), C_6_–C_3_–C_6_ (flavonoids, isoflavonoids), (C_6_–C_3_)_2_ (lignans, neolignans), (C_6_–C_3_–C_6_)_2_ (biflavonoids), (C_6_–C_3_)_n_ (lignins), (C_6_)_n_ (catecholmelanins) and (C_6_–C_3_–C_6_)_n_ (condensed tannins) [62,63,64].

### 4.1. Flavonoids

Flavonoids are one of the largest groups of plant phenols and, by now, more than 8000 structures of flavonoids have been identified. These secondary metabolites are widely distributed in plants and are classified in a number of subgroups, of which one representative of flavones, flavonols, isoflavones, flavanones, flavanonols, flavanols, anthocyanins, chalcones and aurones subgroup is presented in Figure 4. As with other phenols, flavonoids also have numerous functions in plants, such as: Protection against ultraviolet radiation and phytopathogens, a protective response during stress, signaling during development and growth, auxin transport and coloration of flowers for attraction of insects during pollination [65,66].

Apart of being valuable for the plant kingdom, flavonoids are also beneficial to human health. Namely, flavonoids have played a key role in the successful traditional medical treatments of ancient times and their use has continued up to the present day [67,68]. For medicine, the most valuable property of flavonoids is their ability to effectively scavenge highly toxic free radicals and lower oxidative stress [69]. Free radical species occur in the course of numerous physiological processes and can initiate damage of nucleic acid, lipid and protein structures, resulting in disturbance of vital cellular functions and causing a wide range of disorders. Thus, today it is almost impossible to separate free radical reactions and oxidative stress from almost any disorder [70]. Apart from keeping biomolecules safe from free radical attack, flavonoids take part in many biochemical processes in an organism, such as: Regulation of expression of cell cycle regulatory proteins, and inhibition/activation of signal transduction pathways or enzyme activity. As a consequence, flavonoids express many beneficial health actions, such as: Lowering blood pressure and risk of cardiovascular disorders, decreasing the incidence of carcinogenesis and neurodegeneration, inhibiting platelet aggregation and the inflammatory response, as well as lowering levels of bad LDL cholesterol [67,68,69,70,71,72,73,74].

To give answer to question as to why flavonoids express numerous physiological properties is not easy, but the most probable answer lies in the fact that they are highly reactive and can enter into almost any type of reaction known to organic chemistry. Namely, they can take part in oxidation-reduction, acid-base and free radical reactions and hydrophobic interactions, while their substituents can modify electronic induction, resonance and steric hindrance. Additional, flavonoids make stable complexes with metal ions, such as iron, and thus express their antioxidative property, which is the focus of this review [75].

### 4.2. Absorption and Metabolism of Flavonoids in Humans

Absorption and metabolism of flavonoids will be explained with quercetin, as an example, since it is the most abundant flavonoid in human diet. However, other flavonoids follow the same or similar mechanism of absorption and metabolism as described for quercetin.

Quercetin is mainly present in plants in its highly hydrophilic glycosylated forms, mainly as β-glycosides of various sugars. The dominant types of quercetin glycosides differ in plants. However, main forms presented in plants are quercetin-3-*O*-rutinoside (rutin), quercetin-3-*O*-galactoside (hyperoside), quercetin-3-*O*-glucoside (isoquercitrin), quercetin-3-*O*-rhamnoside (quercitrin) and quercetin-4′-*O*-glucoside (spiraeoside) [75].

Prior to absorption in the gut, flavonoids firstly need to be free from plant tissue by chewing in oral cavity and then processed by digestive juices in the intestine or by microorganisms in the colon. Generally, there are two main routes of quercetin glycoside absorption by enterocyte. Firstly, absorption goes via transporter followed by deglycosilation within the enterocyte by cytosolic glycosidase. Secondly, deglycosilation can occur firstly by luminal hydrolases followed by transport of aglycone by passive diffusion or via different transporters. It is demonstrated that quercetin glucosides can be taken up by enterocyte through the sodium–dependent glucose transporter (SGLT1) with subsequent deglycosylation inside the enterocyte by cytosolic β-glycosidase. Also, quercetin glucosides can firstly undergo luminal hydrolysis by lactase phlorizin hydrolase (LPH) and afterwards absorbed inside the enterocyte by passive diffusion or transporter–mediated mechanism [76,77,78,79]. Specifically, quercetin can use glucose transporter (GLUT)-1, -3 and -4 to enter cells and thus operate as an inhibitor of glucose transport [80]. The nature of sugar moiety greatly influences the way and rate of quercetin absorption in the gut. Namely, it is suggested that absorption rate in the small intestine of 3-*O*-glucosylated form of quercetin is higher than the same of quercetin. On the other hand, quercetin glucosides containing rhamnose (rutin) could not be absorbed in the small intestine, and it is believed to be absorbed in the colon after deglycosylation [78,81,82].

The definition of bioavailability states that bioavailability is the portion of an initially administered dose of drug that reaches the systemic circulation unchanged. Considering that, flavonoid bioavailability is very low mostly due to extensive metabolism at the intestinal level. Namely, further biotransformation of quercetin aglycone goes through glucuronidation, sulfonation and methylation of hydroxyl groups, which primarily occurs in enterocytes and hepatocytes. Specifically, major quercetin metabolites detected in plasma are quercetin-3′-sulphate and quercetin-3-glucuronide. It is assumed that they are produced in the small intestine, pass into the portal vein and are further converted into other metabolites in the liver, such as isorhamnetin-3-glucuronide, quercetin diglucuronide, quercetin glucuronide sulphate, methylquercetin diglucuronide, etc. After returning to the bloodstream they are excreted in urine via kidneys. Additionally, a portion of quercetin is converted to low molecular weight phenolic acids, such as 3-hydroxyphenylpropionic acid, 3,4-dihydroxyphenylpropionic acid and 3-methoxy-4-hydroxyphenylpropionic acid [83,84].

### 4.3. Occurrence and Intake of Dietary Flavonoids

Nowadays, a growing body of research confirms different beneficial health effects of dietary flavonoids. Consequently, consumers take more and more interest in the levels and types of flavonoids that are taken up with diet. This is particularly interesting in the scope of the modern concept of functional food, food that apart from nutritional value express additional functions, such as health–promotion or disease prevention. Namely, flavanols and anthocyanidins have been associated with reduction of risk of cardiovascular disease, while anthocyanidins efficiently protect LDL cholesterol oxidation [85]. It had been shown that flavonoids express organ–specificity for cancer prevention, so intake of quercetin rich diet was proven to be in positive correlation with protection against lung and intestinal cancer [86,87].

Flavonoids are present in nearly all edible fruits, vegetables and other food of plant origin. Generally, the human population is consuming notable amounts of flavonoids on a daily basis, being more in regions where diet is mainly based on plant sources. It is estimated that the average daily intake of flavonoids in the United States of America is 20–34 mg, in Finland 24 mg, Japan 63 mg and The Netherlands 73 mg [88]. In Table 4, flavonoids and iron content in selected foods that are regularly consumed in Western diet are listed.

### 4.4. Links between Flavonoids and Iron Homeostasis

Over 30 years ago, it was shown that consumption of tea is in accordance with low non-haem iron bioavailability [90,91]. Consequently, flavonoids, or polyphenols, from the tea were recognized as the main cause for low non-haem iron absorption. Today, flavonoids, among them primarily quercetin, are considered as one of the main dietary inhibitors of iron absorption in the duodenum. Even though the exact mechanism of how flavonoids inhibit non-haem iron absorption is still not fully elucidated, it is strongly believed that its power to chelate iron is mainly responsible for this action [92,93]. In contrast, it was shown that quercetin may operate as a substrate for Dcytb by increasing its reduction potential and providing more Fe^2+^ for cellular uptake by DMT1 [94].

In diseases connected with an imbalance in iron homeostasis, organ-specific iron accumulation is present. In order to bring iron levels back into balance, chelato therapeutics are applied. Potent chelato therapeutics should be able to go through iron-over loaded tissues, complex iron by forming stable and redox-inactive iron and transfer it to Tf in the circulation. Known chelato therapeutic drugs fulfil more or less listed requirements [95]. However, it has been proven in vitro that quercetin is able to decrease intracellular iron and to transfer it to Tf. These significant findings suggest that quercetin could be a valuable representative of chelato therapeutics for iron-redistribution therapy. Yet, this fact still needs to be proven with in vivo studies [96]. On the other hand, it is clear that flavonoids should be avoided in IDA, especially during oral consumption of iron either as a natural constituent of the diet or as a food supplement.

Furthermore, flavonoids were shown to be potent in regulation of systemic iron metabolism. Namely, Bayele et al. [97] reported that quercetin increased expression of hepcidin, a main iron regulatory hormone, which might involve the Nrf2 pathway. Other authors showed in cells that quercetin is able to activate Nrf2 pathway by supporting its nuclear translocation and transcriptional activity [98]. In view of the fact that levels of FPN and H and L ferritin are also known to be transcriptionally up regulated by Nrf2 pathway quercetin could affect iron homeostasis and help cells defending against oxidative stress. Moreover, Vanhees et al. [99] showed that prenatal exposure to quercetin caused hepcidin induction in adult mice.

#### 4.4.1. Flavonoids as Iron Chelators

Flavonoids are known for their numerous health benefits which are mostly attributed to their ability to scavenge highly reactive free radical species. However, flavonoids’ antioxidative potential is, at least partially, associated with their ability to chelate iron. By chelating iron, flavonoids reduce the accessibility of iron to oxygen and consequently diminish oxygen high toxicity, e.g., by inhibiting the production of HO^•^ in Fenton reaction [99].

The exact mechanism by which certain flavonoids reduce bioavailability of non-haem iron is not fully understood, but it is proposed that flavonoids are able to chelate non-haem iron [92,93,100,101,102].

Like most other flavonoids, it was proven that quercetin possesses a high ability to chelate iron [103]. The preferred site for iron chelation by flavonoids, such as quercetin, is its 3-hydroxyl and 4-carbonyl group. Specifically, for complexes containing one iron and one quercetin, the binding strength has an order 3–4 > 4–5 > 3′–4′. Moreover, the 3–4 chelation site is also preferred for complexes which are formed between one iron and two or three quercetin molecules (see Figure 5) [104]. In addition, it is estimated that quercetin, like most other flavonoids, forms a complex with Fe^3+^ with a greater stability than Fe^2+^. Even though when quercetin initially forms a complex with Fe^2+^, Fe^2+^ will autooxidise to Fe^3+^ [105].

Regarding inhibition of iron absorption by quercetin, it was clearly demonstrated in vivo that chelation of iron by the 3-hydroxyl group of quercetin is an important determinate of iron uptake in duodenum [93]. The authors confirmed that the decrease in duodenal iron transfer is due to chelation of iron by quercetin which increases apical uptake of iron, but prevents basolateral transport. Further information that supports this hypothesis is the fact that the quercetin–Fe complex is considerably stable in gastrointestinal conditions. Namely, it was shown in vitro, by mimicking conditions that occur in the stomach, that the recovery of quercetin–Fe complex is up to 45%, which supports the importance of chelation of iron by quercetin in the human body [106]. However, the precise place of iron chelation by quercetin is still uncertain. It is still unknown whether chelation occurs in the duodenal lumen or the cytosol of duodenal enterocytes. One explanation could be that iron is chelated by quercetin in the duodenal lumen by forming the apical–membrane–permeable quercetin–Fe complex that cannot cross the basolateral membrane of enterocyte. Despite its great size, there are in vitro reports that support transport of the quercetin–Fe complex across the cell membrane in both directions [96]. Furthermore, there is evidence that the quercetin–Fe complex is transported by GLUTs transporters [107], which could also be the case in vivo. Furthermore, even though it was shown that quercetin can be transported via GLUTs 1, 3 and 4 transporters, quercetin is lipophilic enough so it can easily cross lipid bilayers without interaction with transporters [80,107,108]. A second possibility is that that quercetin could influx into the cell and then form a complex with free iron. Additionally, it was shown that quercetin may operate as a substrate for Dcytb by increasing its reduction potential and providing more Fe^2+^ for cellular uptake by DMT1 [94]. Knowing this, quercetin could firstly chemically reduce non-haem iron and thus increase apical uptake followed by formation of the quercetin–Fe complex inside the cell. Therefore, both luminal and cytosolic iron chelation, or their combination, can provoke iron accumulation within duodenal mucosa in vivo. However, in both cases the quercetin–Fe complex could remain in the enterocyte due to the inhibition of FPN function or simply because the quercetin–Fe complex would be too bulky to be transported by FPN. This explanation can be applied to all polyphenols that have a noticeable capacity to chelate iron, particularly those which are present in the diet and thus can directly affect iron absorption.

This phenomenon was previously shown for other polyphenols, particularly for (–)-epigallocatechin-3-gallate, but in in vitro conditions using Caco-2 cells as a model system [101,109]. Kim et al. [101,109] reported their finding as unexpected as it was common to think that polyphenols inhibit iron absorption by preventing mainly apical uptake of non-haem iron.

Furthermore, the quercetin-Fe complex inside the cell could be a negative signal for the iron regulatory protein/iron responsive element (IRE/IRP) system and thus destabilize FPN mRNA. Thus, by chelating iron, quercetin could lower free iron levels inside the cell and thus trigger the post-transcriptional IRE/IRP control system, such that when iron levels in tissue are reduced, the expression of FPN is decreased [110]. Furthermore, the possibility that quercetin or its metabolites have direct inhibitory effects on the function of FPN should not be discounted. In addition, bearing in mind that many proteins that have a pivotal role in iron homeostasis beside FPN, such as ferritin, DMT1, TfR1 and Hif-2α, are also regulated by the IRE/IRP system the role of flavonoids which are potent to chelate iron becomes even more important.

Together all these mechanisms could account for the increased mucosal iron retention observed in our studies [93,102]. Furthermore, if a quercetin–Fe complex is formed inside the cell, it could be proposed that quercetin could affect absorption of haem iron too. Namely, quercetin could also prevent the export of free iron for the haem source, after haem degradation by HO-1 which occurs in the cytosol after its absorption.

Potentially listed modes of action could be ascribed to all dietary polyphenols that have demonstrable capacity to chelate iron and this information could be useful in the design of iron chelators based on the structure of these common dietary polyphenols.

In marked contrast, work using Caco-2 cells showed that certain flavonoids promote iron bioavailability (i.e., epicatechin, kaempferol [111,112]). However, these are still in vitro results that needs to be confirmed in vivo.

#### 4.4.2. Flavonoids as Regulators of Systemic Iron Metabolism

It was shown that flavonoids could play important role in regulation of systemic iron metabolism. Namely, a couple of studies have shown that flavonoids have a great effect on hepcidin levels in vivo. However, results are contradictory.

Bayele et al. [97] reported that intraperitoneal quercetin increased hepcidin expression which might involve the Nrf2 pathway, which also correlated with changes in serum iron levels and Tf saturation, as well as with reduction in FPN mRNA. Vanhees et al. [99] showed that prenatal exposure to quercetin caused hepcidin induction in adult mice and the authors hypothesized that after birth, when pups were no longer exposed to quercetin, improved bioavailability of dietary iron sensed as body iron overload. Lesjak et al. [102] showed that quercetin increased hepcidin mRNA levels in both liver and spleen. Increased levels of hepcidin were followed by decreased FPN levels.

However, previous results of others indicate contradictory results on how different dietary polyphenols affect hepcidin levels. Mu et al. [113] reported that the polyphenol myricetin inhibits hepcidin expression induction in vivo by the BMP/SMAD signalling pathway. Quercetin and myricetin are very similar in structure, with myricetin having an extra hydroxyl group. The differences in effect of these two similar polyphenols on iron absorption indicate the complexity of responses to polyphenols. Furthermore, Zhen et al. [114] and Patchen et al. [115] showed that genistein, a main polyphenol from soya, and ipriflavone, synthetic analogue derived from abundant dietary polyphenol daidzein, respectively, both strongly promote hepcidin expression in vivo. Recent studies by Grillo et al. [116] and Zhang et al. [117] indicate that other natural products apart from flavonoids could also have a major role in iron metabolism in vivo and may have potential in therapy of iron metabolism disorders. Namely, Grillo et al. [116] showed that hinokitiol, a natural product of terpenoid structure found in wood that strongly complex iron, could restore iron transport in vivo very effectively. Zhang et al. [117] elucidated that prenylated flavonoid glycoside icariin, which can be found in some Chinese herbal medicinal plants, are able to induce hepcidin expression in mice and change serum and tissue iron concentrations by activating Stat3 and Smad1/5/8 signaling pathways.

Interestingly, it was shown that quercetin directly regulates FPN expression [93]. Namely, in Caco-2 cells exposed to quercetin there was a significant dose-dependent decrease in FPN protein and mRNA and this was associated with a significant decrease in iron transport across Caco-2 cell monolayers. This occurrence was associated with interactions between miRNA and 3′UTR of FPN mRNA. This data suggests a possible great role for miR-17 and potentially other microRNAs in mediating diet-gene interactions that can influence nutrient bioavailability [93].

## 5. Conclusions

All of the above-mentioned possible impacts of flavonoids on iron homeostasis become even more significant in the view that they are consumed regularly in considerable amounts and that nowadays their supplementation is supported due to numerous health benefits. On the other hand, as imbalance in iron homeostasis is connected with many diseases, flavonoids may have important applications in their treatment. Hence it is of great importance to fully understand how dietary flavonoids interact with intake and homeostasis of iron and thus research in this direction should be supported. Further testing of phenolic compounds with iron chelating and cell signaling properties in animal models of iron overload could provide the basis for novel approaches for treating clinical iron deficiency as well as overload in humans. Namely, they could lead to development of new dietary approaches to preventing and treating IDA. In addition, flavonoids might be beneficial for groups at risk of iron loading (e.g., patients with hereditary haemochromatosis), either by limiting the rate of intestinal iron absorption or by modifying tissue iron distribution.

## Figures and Tables

**Figure 1 pharmaceuticals-12-00119-f001:**
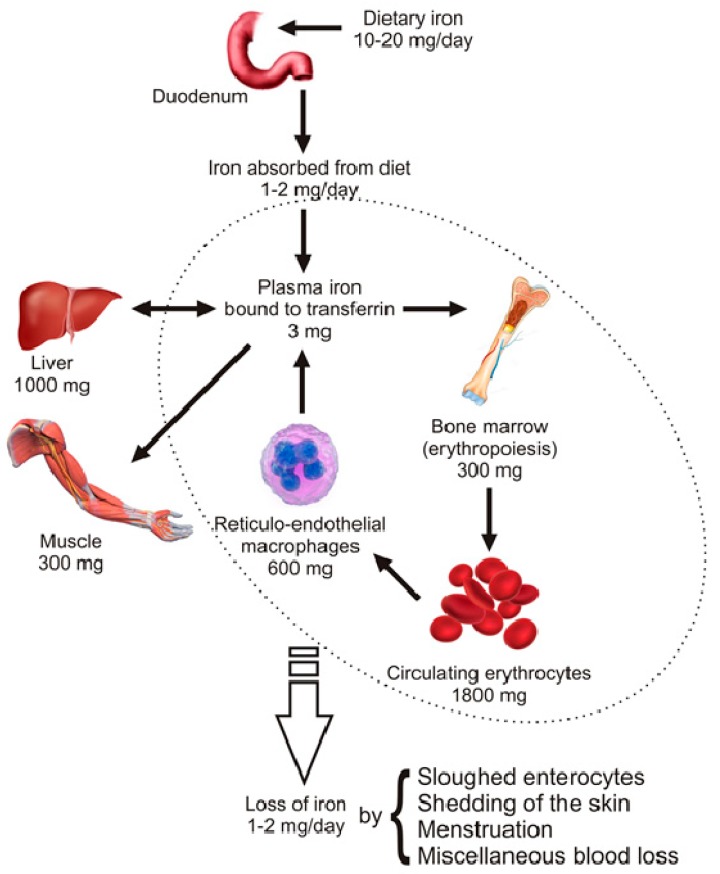
Distribution of body iron. The adult human body has approximately 4 g of iron, with more than half (>2 g) incorporated in the haemoglobin of developing erythroid precursors (300 mg) and mature circulating erytrocytes (1800 mg). The remaining body iron is found in a transit pool in reticulo-endothelial macrophages (600 mg) or stored in hepatocytes (1000 mg). A smaller part is present in muscles within myoglobin (300 mg), while only a minor amount is present in plasma bound to transferrins (Tfs,3 mg) or incorporated in other proteins and enzymes that include iron in their structures. Approximately, 10–20 mg of iron is daily consumed by diet, from which only 1–2 mg is absorbed. The same amount is lost every day by blood loss of different etiology, shedding of the skin and sloughed enterocytes.

**Figure 2 pharmaceuticals-12-00119-f002:**
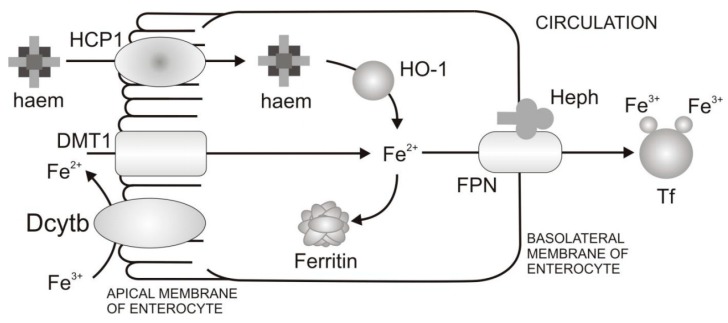
Mechanism of non-haem and haem iron absorption in duodenal cells. Non-haem iron from food is firstly reduced by the ferric reductase Dcytb yielding Fe^2+^, which afterwards enters the enterocytes via divalent metal transporter 1 (DMT1). On the other hand, haem is absorbed via haem carrier protein 1 (HCP1), subsequently broken down by HO-1, after which free Fe^2+^ from haem joins a common cell iron pool with iron from the non-haem source. If body iron stores are high, iron may be stored in the cell complexed with ferritin as Fe^3+^ and eventually lost when the cell is discarded from the intestinal villus tip. Otherwise, iron efflux into the circulation via the iron efflux transporter ferroportin (FPN), subsequently being re-oxidised through hephaestin (Heph) to enable loading into Tf, after which it is transferred to peripheral tissues that require iron.

**Figure 3 pharmaceuticals-12-00119-f003:**
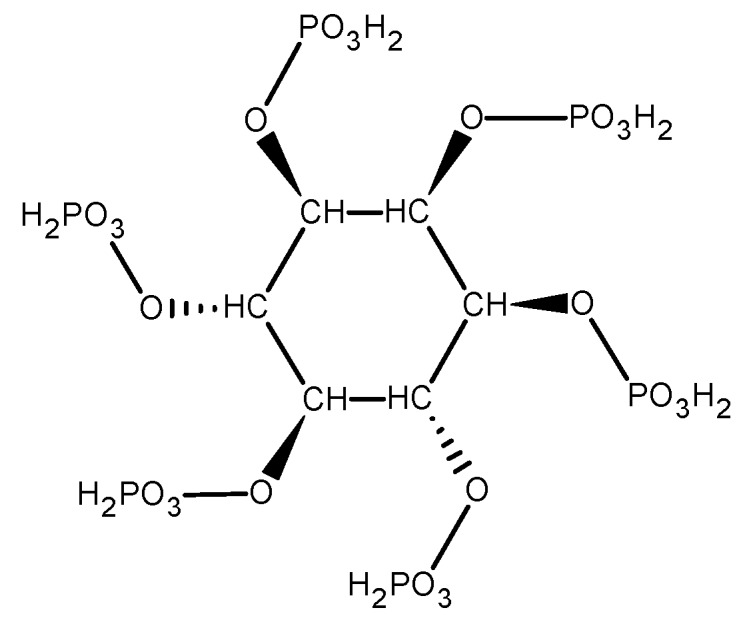
Structure of Phytate.

**Figure 4 pharmaceuticals-12-00119-f004:**
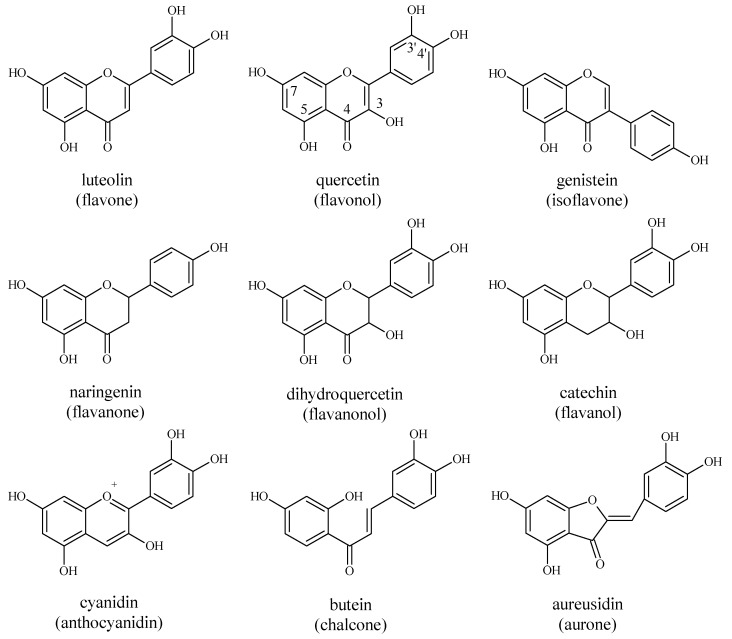
Structure of some classes of flavonoids.

**Figure 5 pharmaceuticals-12-00119-f005:**
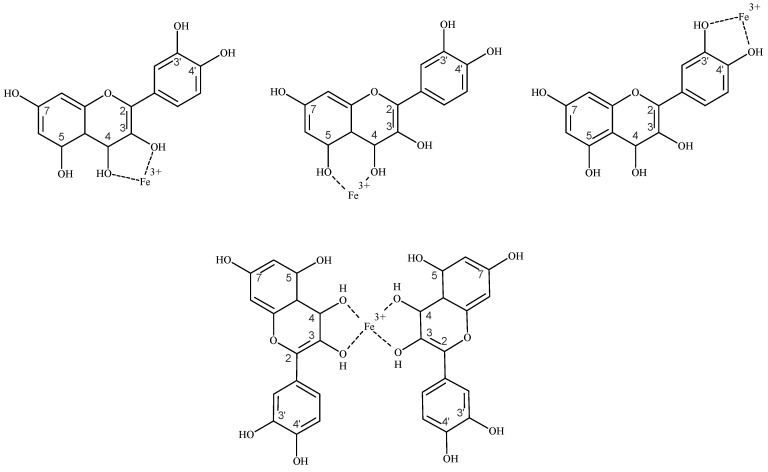
Structures of complexes between Fe^3+^ and quercetin.

**Table 1 pharmaceuticals-12-00119-t001:** Daily requirements of absorbed iron in individuals of different age, sex and state [32,33].

Age/State	Absorbed Iron in Duodenum ^a^ (mg/day)
4–12 months	0.96
13–24 months	0.61
2–5 years	0.70
6–11 years	1.17
12–16 years (girls)	2.02
12–16 years (boys)	1.82
Adult males	1.14
Women during lactation	1.31
Women during menstruating period	2.38
Women during postmenopausal period	0.96
Women 1st trimester of pregnancy	0.8
Women 2nd & 3rd trimester of pregnancy	6.3

^a^ Calculations were done on the basis of average weight and average status.

**Table 2 pharmaceuticals-12-00119-t002:** Amount of total iron in common foods [39].

Food	mg Iron/100 g Food
sources of non–haem iron	
red bean	6.69
parsley	6.20
wheat flour, whole-grain	3.71
corn flour, whole-grain, yellow	2.38
garlic	1.70
lettuce	0.86
potato	0.81
orange	0.80
red cabbage	0.80
broccoli	0.73
blackberry	0.62
kiwi	0.54
red pepper	0.43
cauliflower	0.42
strawberry	0.41
apricot	0.39
fig	0.37
carrot	0.30
cucumber	0.28
blueberry	0.28
banana	0.26
watermelon	0.24
eggplant	0.23
red onion	0.21
apple	0.12
sources of haem iron	
goose, liver	30.53
oyster	3.86
beef meat	1.69
lamb meat	1.55
turkey meat	1.09
chicken meat	0.82

**Table 3 pharmaceuticals-12-00119-t003:** Phytate and iron content in selected foods [39,43].

Food	g Phytate/100 g Food	mg Iron/100 g Food
soybean seed	1.0–2.22	15.7
sesame seed	1.44–5.36	14.5
bean	0.61–2.38	9.0
lentil	0.27–1.51	7.4
flax seed	2.15–3.69	7.2
indian walnut	0.19–4.98	6.7
sunflower seeds	3.9–4.3	6.0
wheat seed	0.39–1.35	5.3
oats	0.42–1.16	4.7
pea	0.22–1.22	4.7
hazelnut	0.23–0.92	4.7
peanut	0.17–4.47	4.5
chickpeas	0.28–1.60	4.3
rice	0.06–1.08	4.0
pistachio nuts	0.29–2.83	3.9
almond nuts	0.35–9.42	3.7
corn	0.72–2.22	3.0
walnut	0.20–6.69	2.9
rye seed	0.54–1.46	2.6

**Table 4 pharmaceuticals-12-00119-t004:** Flavonoids and iron content of selected foods [39,89].

Food	mg Flavonoid/100 g	mg Iron/100 g
parsley	233.16	6.20
garlic	3.61	1.70
lettuce	4.63	0.86
red cabbage	210.67	0.80
broccoli	11.96	0.73
red pepper	0.86	0.43
cauliflower	1.02	0.42
strawberry	13.35	0.41
fig	8.07	0.37
carrot	0.60	0.30
blueberry	180.82	0.28
cucumber	0.17	0.28
tomato	5.95	0.27
banana	13.69	0.26
cranberry	132.08	0.23
eggplant	85.73	0.23
red onion	56.61	0.21
apple	15.15	0.12

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
