# Peer review of "Role of Dietary Flavonoids in Iron Homeostasis"

_pharmaceuticals, 2019, doi:10.3390/ph12030119_

Round 1

Reviewer 1 Report

The authors present a very detailed review of the perceived effects and benefits of flavonoids specifically on iron homeostasis, there are a lot of potential mechanisms that are alluded to where flavonoids can be affecting iron homeostasis, it would be good if these could be summarised into a figure somehow. The other comment/question that I have is, the authors have based the explanation on quercetin and used this as a general model to explain how flavonoids have been suggested to play a role in iron homeostasis, does this hold true for other flavonoids? As they have already mentioned in the para starting in Line 501 that there are studies which suggest otherwise, it would be important for the reader if the authors summarise these finding as well.

Other comments

Line 115- Although the authors mention that there is not much known about intracellular iron transport, they should discuss the role for NCOA4 and PCBP proteins with respect to this function briefly.

Line 483- the authors provide an extensive overview and review of the use of flavonoids in decreasing iron, but apart from the mention in lines 483-485 they fail to discuss the importance of studies showing other flavonoids which have been shown to increase iron availability, was this due to the models used? Or was it due to structural differences?

Line 501- is there only one study which suggests flavonoids can decrease hepcidin expression, all the other studies mentioned in this paragraph show an increase in hepcidin synthesis on treatment with flavonoids, please elaborate on Grillo et al 2017 and Zhang et al 2016 and how these studies have added to our knowledge of the effect of flavonoids on iron homeostasis in addition to the fact that the compounds used in these studies were also potential iron chelators.

Minor comments:

Line 150: error in table legend- please fix

Line 183 is a repeat of line 160.

Line 190- to keep iron homeostasis in balance

Page 7 is empty- maybe due to formatting when converted into PDF- please check

 Line 224- mainly for prevention of

Line 337-hydrophilic

Line 353- greatly influences the way

Line 354-356 is unclear, please rephrase.

Line 372-376- please check language- a growing body of evidence….. scientific community is taking more interest…. Concept of so called…..

Line 464 typos

Line 502- remove thus

Author Response

Reviewer 1 Comments

The authors present a very detailed review of the perceived effects and benefits of flavonoids specifically on iron homeostasis, there are a lot of potential mechanisms that are alluded to where flavonoids can be affecting iron homeostasis, it would be good if these could be summarised into a figure somehow.

The other comment/question that I have is, the authors have based the explanation on quercetin and used this as a general model to explain how flavonoids have been suggested to play a role in iron homeostasis, does this hold true for other flavonoids?

According to authors flavonoids can affect iron metabolism by two mechanisms – chelating iron and changing expression of genes/proteins important in iron metabolism. For chelation, it could be assumed that all flavonoids which have favourable hydroxyl groups for iron chelation could affect iron absorption as quercetin. However, the same assumption cannot be made for changing expression of genes/proteins as it was shown that flavonoids which have similar structure (quercetin and myricetin) change iron metabolism in different ways.

As they have already mentioned in the para starting in Line 501 that there are studies which suggest otherwise, it would be important for the reader if the authors summarise these finding as well.

All findings regarding how other polyphenols affects systemic iron metabolism in vivo are already summarised in the paragraph starting in line 501. There is no additional scientific paper about this issue known to authors.

Other comments

Line 115 Although the authors mention that there is not much known about intracellular iron transport, they should discuss the role for NCOA4 and PCBP proteins with respect to this function briefly.

Role of NCOA4 and PCBPs are added into text as well as corresponding reference.

Line 483 the authors provide an extensive overview and review of the use of flavonoids in decreasing iron, but apart from the mention in lines 483-485 they fail to discuss the importance of studies showing other flavonoids which have been shown to increase iron availability, was this due to the models used? Or was it due to structural differences?

It was due to models used. Listed references only proven those findings in vitro using cells.

Line 501 is there only one study which suggests flavonoids can decrease hepcidin expression, all the other studies mentioned in this paragraph show an increase in hepcidin synthesis on treatment with flavonoids, please elaborate on Grillo et al 2017 and Zhang et al 2016 and how these studies have added to our knowledge of the effect of flavonoids on iron homeostasis in addition to the fact that the compounds used in these studies were also potential iron chelators.

According to authors knowledge yes there is only one study which shows decrease in hepcidin expression by polyphenols in vivo. While other mentioned studies in the paragraph are only references known by authors which show increase of hepcidin expression by polyphenols, or polyphenols like compounds in vivo, which are different from quercetin.

The additional info on Grillo et al 2017 and Zhang et al 2016 results are added.

Minor comments:

Line 150 error in table legend- please fix

This error was due to formatting by system during submission. In original word document blank page do not exists.

Line 183 is a repeat of line 160.

Repetition in line 183 is deleted.

Line 190 to keep iron homeostasis in balance

Corrected.

Page 7 is empty - maybe due to formatting when converted into PDF - please check

Yes, it was due to formatting by system during submission. In original word document blank page do not exists.

Line 224 mainly for prevention of

Corrected.

Line 337 hydrophilic

Corrected.

Line 353 greatly influences the way

Corrected.

Line 354-356 is unclear, please rephrase.

Specified lines are rephrased so to be clearer.

Line 372-376 please check language - a growing body of evidence….. scientific community is taking more interest…. Concept of so called…..

Specified language is corrected.

Line 464 typos

Authors are not sure on what typos (mistakes) reviewer thinks? We would be grateful if reviewer specify them.

Line 502 remove thus

World Thus is removed.

Reviewer 2 Report

The manuscript describes a potentially interesting topics; however, the use of references is inappropriate and the tables are misleading.

As to references, the manuscript often includes strong statements without an appropriate reference. Example: Line 53: "40 % of all maternal prenatal deaths are linked to anemia". Worldwide?  Locally? What about infection, haemorrhage? Such a statement is useless without a reference.

From page 15, the manuscript switches from numbered references to the first author reference, which makes looking up of the references almost impossible. 

As to tables: Spinach (a notorious case in iron history) contains 260 ppm of iron according to Table 1, and 27 ppm according to Table 4.

Overall, the manuscript presents a lot of data; however, it often uses very broad general terms (lines 310-325). Conclusions are missing and there is not a clear recommendation as to which food should be avoided/recommended in which condition.  

Author Response

Reviewer 2 Comments and Suggestions

The manuscript describes a potentially interesting topics; however, the use of references is inappropriate and the tables are misleading.

References and tables are corrected.

As to references, the manuscript often includes strong statements without an appropriate reference. Example: Line 53: "40 % of all maternal prenatal deaths are linked to anemia". Worldwide? Locally? What about infection, haemorrhage? Such a statement is useless without a reference.

It is a worldwide phenomenon and reference is listed. It is reference under number 6 (World Health Organization Iron deficiency anaemia: assessment, prevention, and control. A guide for programme managers; WHO Press: Geneva, Switzerland, 2001. Page 9). Most of references are put on the end of each paragraph, not behind each sentence, since most sentences are mix of facts from different sources.

From page 15, the manuscript switches from numbered references to the first author reference, which makes looking up of the references almost impossible.

In revised review all references are changed to numbered references.

As to tables: Spinach (a notorious case in iron history) contains 260 ppm of iron according to Table 1, and 27 ppm according to Table 4.

Data in tables are corrected so that same food have same amounts of iron. Data for spinach are erased so not to cause confusion, as nobody is sure what is exact amount of iron in spinach. Data for all tables are extracted from relevant source (USDA Food Composition Databases. Available online: https://ndb.nal.usda.gov/ndb/search/list?home=true).

Overall, the manuscript presents a lot of data; however, it often uses very broad general terms (lines 310-325).

Authors agree. However, for the specified paragraph, we wanted to give general overview of flavonoids health benefits, with not going into details, since it is not a focus of review, and referencing relevant literature if reader is interested to know more.

Conclusions are missing and there is not a clear recommendation as to which food should be avoided/recommended in which condition.

The authors wanted to point out that flavonoids are very promising target for treating various disorders of iron homeostasis. Also, authors wanted to emphasise that more research in the field needs to carried out, since this field of research is seemed neglected. Thus, at the moment there are no sufficient data to recommend which particular food should be avoided/recommended in which condition.

Round 2

Reviewer 2 Report

The manuscript describes potentially very interesting topics; however, it suffers from making too many statements without appropriate references. Since the manuscript represents a Review, the lack of addressed references provides a major setback.

Most striking examples from the iron field metabolism are:

Line 52: “iron deficiency greatly reduces resistance to infection” – this is as a general statement difficult to understand without an addressed  reference. There is some evidence that iron deficiency actually protects against malaria (Gwamaka et al., Iron deficiency protects against severe Plasmodium falciparum malaria and death in young children. Clin Infect Dis. 2012 Apr;54(8):1137-44).

The effect of iron on M. tuberculosis has been recently reviewed in Pharmaceuticals (Agoro R, Mura C. Iron Supplementation Therapy, A Friend and Foe of Mycobacterial Infections? Pharmaceuticals (Basel). 2019 May 17;12(2); this reference shows a potential negative effect of iron supplementation.

Line 73: About 52% of total body is found as part of hemoglobin... Without a reference, it is impossible how the authors reached this (very exact) quantification. Actually, one of the two references cited at the end of paragraph (Andrews et al. 1999) states that “more than two thirds of of the body iron content is incorporated in hemoglobin”.

Figure 2: The role of HCP1 in heme transport is rather controversial, therefore, HCP1 should not be included in the Figure without adequate explanation.

Line 173-177: The authors fail to mention chronic bleeding as one of the main causes of anemia.

Line 315-330: There are many key statements in this paragraph; therefore, two references to review papers are grossly inadequate.

Line 403: Sentence is impossible to understand without a reference. Do the authors suggest that iron deficiency actually leads to iron accumulation in some organ?

Line 501-502: The authors discuss a known relationship between hepcidin and FPN protein levels (reference 103); however, the discussed paper (reference 92) shows Fpn mRNA, and is therefore not compatible with reference 103.

Overall, the manuscript presents a lot of statements and conclusions, which are difficult to interpret without addressed references. One of the purposes of a review article is to summarize the current knowledge in the field for a newcomer, and to point out the most important original papers. It is felt that the manuscript is grossly lacking in this respect.

Author Response

Reviewer 2 Comments

The manuscript describes potentially very interesting topics; however, it suffers from making too many statements without appropriate references. Since the manuscript represents a Review, the lack of addressed references provides a major setback.

Most striking examples from the iron field metabolism are:

Line 52: “iron deficiency greatly reduces resistance to infection” – this is as a general statement difficult to understand without an addressed reference.

There is some evidence that iron deficiency actually protects against malaria (Gwamaka et al., Iron deficiency protects against severe Plasmodium falciparum malaria and death in young children. Clin Infect Dis. 2012 Apr;54(8):1137-44).

The effect of iron on M. tuberculosis has been recently reviewed in Pharmaceuticals (Agoro R, Mura C. Iron Supplementation Therapy, A Friend and Foe of Mycobacterial Infections? Pharmaceuticals (Basel). 2019 May 17;12(2); this reference shows a potential negative effect of iron supplementation.

The reviewer is absolutely right. The statement is too general and may easily lead the reader to confusion. Thus, the statement “iron deficiency greatly reduces resistance to infection” is erased and two new references are put at the end of the paragraph, which stress importance of iron in development of microbial pathogens infection and innate immunity.

Line 73: About 52% of total body iron is found as part of hemoglobin... Without a reference, it is impossible how the authors reached this (very exact) quantification. Actually, one of the two references cited at the end of paragraph (Andrews et al. 1999) states that “more than two thirds of the body iron content is incorporated in hemoglobin”.

The 52% percentage is simply calculated using amounts of iron in different parts of the body presented in Figure 1, which is for iron found as part of haemoglobin in circulating erythrocytes or erythrocyte precursors exactly 52.2%. This is based on iron in circulating erythroid precursors and mature red blood cells. We had made a mistake, since we had failed to add iron in macrophages which is being released from engulfed red blood cells. Taking this into account haemoglobin iron content represents 67.5% of body iron which is more than two third of total body iron levels.

We have changed the statement to take this into account

The statement reads:

About 66% of total body iron is found as part of haemoglobin in circulating erythrocytes, erythrocyte precursors or as intracellular pool (liver and reticulo-endothelial macrophages), 7.5% in muscle as part of myoglobin, 0.5% as part of the catalytic center of a variety of enzymes (cytochromes, catalase, peroxidases, flavoproteins, etc.) and 0.1% as Tf-bound iron in the circulation (see Figure 1)

Figure 2: The role of HCP1 in heme transport is rather controversial, therefore, HCP1 should not be included in the Figure without adequate explanation.

Controversial role of HCP1 is explained and appropriate references added.

Line 173-177: The authors fail to mention chronic bleeding as one of the main causes of anemia.

Chronic bleeding, as one of causes of IDA, is added.

Line 315-330: There are many key statements in this paragraph; therefore, two references to review papers are grossly inadequate.

Relevant references are added.

Line 403: Sentence is impossible to understand without a reference. Do the authors suggest that iron deficiency actually leads to iron accumulation in some organ?

No. Authors wanted to say that accumulation of iron in some organs can be present during anaemia also, which is paradoxically, referring to diseases such as atransferrinaemia or β-thalassaemia. However, in order to avoid confusion among readers part of the sentence mentioning anaemia is erased.

In addition, it is noted that anaemia of inflammation results in iron accumulation in some organs such as spleen and liver (Nemeth, E.; Ganz, T. Anemia of inflammation. Hematol Oncol Clin North Am. 2014 28: 671-681).

Line 501-502: The authors discuss a known relationship between hepcidin and FPN protein levels (reference 103); however, the discussed paper (reference 92) shows Fpn mRNA, and is therefore not compatible with reference 103.

Sentence refereeing to reference 103 is erased. We have not published this yet but levels of hepcidin protein go up after quercetin treatment and ferroportin levels go down.

Overall, the manuscript presents a lot of statements and conclusions, which are difficult to interpret without addressed references. One of the purposes of a review article is to summarize the current knowledge in the field for a newcomer, and to point out the most important original papers. It is felt that the manuscript is grossly lacking in this respect.